# Perceptions and Intentions around Uptake of the COVID-19 Vaccination among Older People: A Mixed-Methods Study in Phuket Province, Thailand

**DOI:** 10.3390/ijerph20115919

**Published:** 2023-05-23

**Authors:** Chayanit Luevanich, Ros Kane, Aimon Naklong, Prapaipim Surachetkomson

**Affiliations:** 1Public Health Program, Faculty of Science and Technology, Phuket Rajabhat University, Phuket 83000, Thailand; 2School of Health and Social Care, University of Lincoln, Brayford Pool, Lincoln LN6 7TS, UK; 3Teaching Profession Department, Faculty of Education, Phuket Rajabhat University, Phuket 83000, Thailand; 4Science and Mathematics Program, Faculty of Science and Technology, Phuket Rajabhat University, Phuket 83000, Thailand

**Keywords:** COVID-19 vaccination, perception, intention, decisions to uptake vaccine

## Abstract

Background: A 70% vaccination rate against COVID-19 in the general population was required for re-opening Phuket tourist industry. However, prior to this research, 39.61% of older people remained unvaccinated. This study aimed to examine perceptions and intentions around COVID-19 vaccination amongst older people and to explore the reasons and factors influencing their decisions to receive or refuse vaccination. Methods: This was a mixed-methods approach with a sequential explanatory design. We conducted an online survey and semi-structured qualitative interview with a subsample. Multinomial logistic regression was applied and thematic content analysis was conducted. Results: 92.4% of participants reported intention to receive the vaccine. Multinomial regression analysis revealed that perceived barriers (AdjOR = 0.032; 95% CI: 0.17–0.59), perceived benefit (AdjOR = 2.65; 95% CI: 1.49–4.71), good health (AdjOR = 3.51; 95% CI: 1.01–12.12) and health not good (AdjOR = 0.10; 95% CI: 0.02–0.49) were predictors of vaccine uptake. In the qualitative interviews, four key influences on up-take for the 28 vaccinated participants were: prevention and protection, convenience, fear of death from COVID-19, and trust in the vaccine. Four key influences on refusal of vaccination in the eight unvaccinated participants were: rarely leaving the house, fear of vaccine side-effects, fear of death after getting the vaccine, and not enough information for decision-making. Conclusion: Intervention and campaigns addressing COVID-19 vaccination should employ strategies, including the widespread use of social and other popular media to increase older people’s perceived benefit of vaccination on their current and future health status, while decreasing perceived barriers to receiving the vaccine.

## 1. Introduction

From early on in the COVID-19 pandemic, vaccines, along with the wearing of face masks, social distancing, and advice around hand washing, were central public health interventions to control the spread of infection and contribute to achieving herd immunity and reducing the risk of severe illness [1]. During the pandemic, as new vaccines emerged, the WHO authorized a number of vaccines for emergency use globally, including Oxford-AstraZeneca, Johnson & Johnson, Pfizer-BioNTech, Moderna, Sputnik V, and Sinovac. This swift response resulted in some concern about vaccine safety, efficacy [2,3], side effects, effectiveness [4], longer-term side effects [5], and post-vaccination complications. In some cases, this led to refusal or hesitancy to consent to vaccination [6].

Phuket province, which is a popular tourist destination, was one of the first provinces in Thailand to confirm COVID-19 cases, and the pandemic had a serious financial impact, including the total shutdown of the tourist industry. In response, the Phuket tourism industry network and the Thai government introduced a new “Phuket sandbox scheme” to help business recovery through the re-opening of the tourism industry. One requirement for the receipt of the sandbox scheme was that Phuket had to achieve a COVID-19 population vaccination rate of 70%, and this provided an economic incentive to maximize vaccination uptake. Phuket was, therefore, the first Thai province to implement a free COVID-19 vaccine program whereby vaccination was offered initially to healthcare providers (in March 2021) and subsequently to the general population (on 1April 2021) and older people (on 5 June 2021). During the initial phase of the vaccine roll-out, the Thai government provided Sinovac for 18–59-year-olds and Oxford-AstraZeneca for older people. Initially, vaccination rates were low: in 2021, 39.61% of older people in Phuket infected with COVID-19 were unvaccinated, and of those over age 60 who died from the infection, 9.35% were unvaccinated [7]. Older people are a key group that the WHO recommends for COVID-19 vaccination due to their potentially higher levels of underlying medical conditions and because the severity of illness from COVID-19 increases with age [8], especially among those with co-morbidities [9,10]. Moreover, full vaccination can prevent hospitalization among older people [11], and so there was a pressing need to better understand the motivations of older people around vaccine uptake.

Early studies showed that, in Thailand, 44.3% of older Thai people were hesitant to receive the COVID-19 vaccination [12]. This compares with 74% in Hong Kong, 56.15% in Saudi Arabia, and 8.7% in the USA [13,14]. People’s intention or hesitancy to receive a vaccine is likely to be based on different social contexts, cultures, and policies, and it is important to understand these influencing factors to better understand how to overcome vaccine hesitancy and, in turn, reduce the severity of symptoms and risk of hospitalization. Although recent research has investigated vaccine hesitancy among older populations [13,15,16], there was a specific need to conduct this study in Phuket, as it was the first Thai province to introduce vaccination and the economic imperative to achieve 70% coverage to qualify for the sandbox recovery scheme. The aims of the study were therefore (1) to examine perceptions and intentions around COVID-19 vaccination amongst older people in Phuket and (2) to explore the reasons and factors influencing their decisions to receive or refuse vaccination.

## 2. Materials and Methods

### 2.1. Study Design

This was a mixed-methods approach with a sequential explanatory design [17] to gain a deep understanding of the complex determinants of decision-making around COVID-19 vaccine uptake in the older Phuket population. An online cross-sectional survey was used to determine the population’s perceptions and intentions, and qualitative, semi-structured individual interviews were conducted to better understand the different views on COVID-19 vaccine acceptability and refusal and to explore the reasons behind these views.

### 2.2. Quantitative Component

The quantitative component included an anonymous cross-sectional online self-reported questionnaire to collect data from the population of older adults (aged 60+) dwelling in Phuket Province, Thailand.

#### 2.2.1. Questionnaire Design and Piloting

The questionnaire was developed in the Thai language by the research team following a comprehensive review of the relevant literature. The first part of the questionnaire collected participants’ demographic data (age, sex, education, occupation, medical history, health status, intention to uptake the COVID-19 vaccine, source of information, and history of vaccination). The second part focused on the participants’ perception of the COVID-19 vaccine and was based on the health belief model (HBM) [18], which has been widely used for examining health behavior, health service, and the context of vaccination [19] and is useful for studying COVID-19 hesitancy [20]. The HBM was used to assess participants’ perceived susceptibility, benefits, access barriers, and cues to action relating to vaccine uptake. The 32 item questions were constructed around a 5-point Likert scale (strongly agree = 5, agree = 4, neutral = 3, disagree = 2, strongly disagree = 1). The content validity was determined based on the index of item objective congruence (IOC) by three experts. A pilot sample (n = 30) was used to improve the language and test reliability using Cronbach’s alpha coefficient, which was 0.785. Sub-items of perception of the COVID-19 vaccine with Cronbach’s alpha coefficient were as follow: perceived susceptibility: 0.804, perceived severity: 0.832, perceived benefits: 0.895, perceived access barriers: 0.712, and cues to action: 0.895.

#### 2.2.2. Data Collection

The survey was conducted during the first phase of the COVID-19 vaccination roll-out in Phuket (from 1 June to 20 October 2021). An online social media strategy was adopted whereby invitations to take part in the research were sent out via professional and individual networks. The link to the questionnaire and other relevant study documentation was disseminated. A participant information sheet explained the research aims and the inclusion criteria (for the survey component of the study, being aged 60 or over and a Thai resident in Phuket) and that participation in the study was voluntary. A statement assuring participants of confidentiality was also included and the consent form was embedded within the link. We sent the link electronically to our networks and colleagues and requested that they distribute the survey link, targeting those who met our inclusion criteria. To promote uptake, members of the research team also had some physical presence onsite at one Tambon Health Promoting Hospital older people’s clinic where face-to-face data collection also took place. To those consenting to take part, researchers read each question directly to individuals and participants selected the appropriate answers.

The sample size was calculated based on a 95% confidence level and a 0.05 margin of error, assuming an observed proportion of respondents selecting a specific response option of 50% and finite correction for the study population of 33,080 older people. The calculated sample size was 380 older people. This study considered an additional 5% to account for the incomplete survey, making the final sample size 406.

#### 2.2.3. Data Analysis

Descriptive statistics (frequencies and percentages) were used to summarize all variables and, because the measurement variable did not meet the normality assumption of a one-way ANOVA [21], a Kruskal-Wallis test was used to compare perceptions of the COVID-19 vaccine among participants who had received a vaccine, were unsure, or had not been vaccinated. A pairwise comparison of the perception of the COVID-19 vaccine by a post-hoc test was conducted. Multinomial logistic regression analyses were used to identify the predictors of vaccine intention with adjusted odds ratios (ORs) and 95% confidence intervals (CIs). The main criteria for entering the variables in the regression model was a *p*-value < 0.05. The level of significance was set at 0.05.

### 2.3. Qualitative Component

The qualitative study used inductive in-depth interviews to explore contextual influences on decision-making around vaccine uptake. On completing the survey, participants were asked to provide their contact details if interested in taking part in a follow-up interview. Eight participants indicated on the questionnaire that they were willing to be re-contacted and the research team subsequently managed to contact six participants who later took part in individual interviews. The remaining thirty who were interviewed were recruited directly following face-to-face interaction whilst completing the questionnaire at the hospital clinic.

The interview topic guide was informed by a literature review and included participants’ perceptions of the COVID-19 pandemic and the reasons why they had received or refused the COVID-19 vaccine. The open-ended questions were: What is your opinion on the type of COVID-19 vaccine that the government provides for older people, and why? Why did you decide to get vaccinated against COVID-19? Did you decide to get the COVID-19 vaccine immediately? How? Why? Why are you not getting vaccinated against COVID-19? What informed that decision? Informed consent was obtained before the in-depth interview commenced.

We conducted all interviews face to face and kept a distance of at least 1 m from each participant. A purposive sample of thirty-six participants who met the inclusion criteria for this component (Thai residents of Phuket who were sixty years old and over without physical, cognitive, or hearing impairment, who agreed to take part in the interview and were willing to sign the consent form) was invited to take part in the quantitative component of the study.

The interviews took place from 1 August to 30 November 2021, and all were conducted in the Southern Thai language and audio recorded. The duration of the interviews was 30–45 min. No financial incentives were offered to the participants. The audio recordings were transcribed verbatim after each interview. Thematic analysis was employed in this study using the stages outlined: data familiarization, coding, and theme identification and refinement [22]. The transcribed interviews were read and coded by the first author. To enhance the rigor of the analysis, coding approaches and subsequent theme generation and refinement were discussed between the first author and the other researchers. All analysis was conducted in Thai and the results were translated into English for publication and wider dissemination.

## 3. Results

Four hundred and six participants completed the questionnaire. Two hundred and seventy- eight (68.5%) were female, and the average age of participants was 68.8 years (SD = 6.6) (see full participant characteristics in Table 1). Overall, 92.4% of participants indicated their intentions to receive the COVID-19 vaccine (or had already received the vaccine), 3.2% expressed a refusal of vaccination, and 4.4% indicated that they were not sure.

Table 1 shows that most of the demographic variables were not significantly associated with vaccination decisions. However, one variable was significantly related to the intention to be vaccinated: perceived overall health status.

Table 2 presents the participants’ perception of the COVID-19 vaccine in relation to their decision around vaccination; the results showed that participants’ perceptions of the COVID-19 vaccine in perceived severity, perceived barriers, cues to action, and perceived benefits, were statistically significantly different between the ‘intention’ group, ‘no intention group’, and ‘not sure’ group. Perceived severity ($X^{2}$ = 6.3, *p* = 0.043) had a mean rank score of 205.2 for the ‘intention’ group, of 126.7 for the ‘no intention’ group, and of 224.0 for the ‘not sure’ group. Perceived barriers ($X^{2}$ = 18.3, *p* < 0.001) had a mean rank score of 196.8 for the ‘intention’ group, of 253.3 for the ‘no intention’ group, and of 307.8 for the ‘not sure’ group. Cues to action ($X^{2}$ = 39.0, *p* < 0.001) had a mean rank score of 213.3 for the ‘intention’ group, of 32.4 for the ‘no intention’ group, and of 122.1 for the ‘not sure’ group. Perceived benefits ($X^{2}$ = 31.8, *p* < 0.001) had a mean rank score of 212.8 for the ‘intention’ group, of 77.4 for the ‘no intention’ group, and of 101.4 for the ‘not sure’ group.

Table 3 shows the results from the post-hoc test of the mean rank of perceived barriers. The adjusted significance of the ‘No intention-Intention’ group was statistically significant at 0.05 (*p* < 0.001). The adjusted significance of post-hoc tests of the mean rank of cues to action were statistically significant at 0.05 (*p* < 0.001) for the ‘No intention-Intention’ group and at 0.05 (*p* = 0.004) and 0.05 (*p* < 0.001) for the ‘Not sure-Intention’ group. The adjusted significance of post-hoc tests of the mean rank of perceived benefits for the ‘No intention-Intention’ and ‘Not sure-Intention’ groups had statistically significant differences at 0.05 (*p* < 0.001).

Table 4 shows the results of the multinomial logistic regression model, comparing the group of participants who indicated no intention to receive the COVID-19 vaccine (reference category), participants who indicated an intention, and those who were unsure whether or not to receive the COVID-19 vaccine. The variables which can predict the intention to uptake the COVID-19 vaccine among older people in Phuket Province, when compared with those with no intention to receive a vaccine, were ‘perceived barriers’ (AdjOR = 0.32; 95% CI: 0.17–0.59), ‘perceived benefits’ (AdjOR = 2.65; 95% CI: 1.49–4.71), and interestingly, both ‘good health’ (AdjOR =3.51; 95% CI: 1.01–12.12) and ‘not good health’ (AdjOR = 0.10; 95% CI: 0.02–0.49) (moderate health status was the reference category). The variable which can predict being unsure, compared with having no intention to receive a vaccine, was good health status (AdjOR =5.47; 95% CI: 1.02–29.30).

The interviews explored the reasons and factors influencing the decisions of the sub-group of thirty-six older people to receive or refuse vaccination against COVID-19. Twenty-eight of the participants interviewed had already been vaccinated at the point of their interview. Most of the interview participants were women (n = 24). The average age was 70.3 (S.D. = 8.3); twenty-five had long-term conditions, thirteen of whom had more than one. Further details and presented in Table 5. Most had been made aware of the campaign for vaccination from television and online groups.

Four key influences for COVID-19 vaccine uptake were identified from those vaccinated and are detailed in turn below.

### 3.1. Prevention and Protection

For some participants, the main motivations for vaccination were prevention and protection. Because the research took place during the height of the pandemic, many new and cumulative cases were being reported daily. Participants thought that the vaccine could prevent infection and/or protect against COVID-19, particularly against serious illness or complications, and increase their immunity so that they might combat COVID-19 if infected: *“Better to inject than not inject. If I inject, I will have immunity. If I have a few symptoms, I can tolerate it.” (Male, 79 years old, colon cancer and benign prostatic hyperplasia).*

### 3.2. Convenience

During the height of the pandemic, the Thai government introduced travel restrictions within the country, and this provided motivation for some to be vaccinated. Those wishing to travel to another province had to show a COVID-19 vaccine certificate. *“I think to get vaccination is better than to not receive it. If you don’t get the vaccine, you can’t go anywhere. I also like to travel. If you do get the vaccine, you can visit other provinces.” (Female, 79 years old, diabetes mellitus and hypertension)*

### 3.3. Fear of Death from COVID-19

Three of the twenty-eight vaccinated interview participants reflected that COVID-19 is the main cause of death in older people and recognized the importance of the vaccine in protecting them from COVID-19-related illness and death. *“I don’t want to die. I am a village health volunteer. I risk dying from COVID-19. So I get the COVID-19 vaccine.” (Female, 62 years old, no underlying disease)*

### 3.4. Trust in the Vaccine

Older people who decided to be vaccinated had trust in the efficacy and effectiveness of the vaccine and confidence in the vaccine, possibly influenced by its endorsement by the government and the medical profession. *“The government and doctors selected the vaccine for the Thai people. I think they chose a good vaccine. If it is not good, they lost their name. They had to find the fastest vaccine available to Thai people at that time.” (Female, 77 years old, hypertension and dyslipidemia).*


*Responses from those who refused vaccination*


Eight interview participants (mostly women, n = 7) had refused the COVID-19 vaccine. The average age was 75.3 (S.D. = 5.8); six had chronic diseases. The unvaccinated interview participants gave four main reasons for their refusal:


*Rarely leaving the house*


Six of the eight participants reported how, during the COVID-19 pandemic, they rarely went out of their homes, so felt their chances of being exposed to the infection were low and vaccination, therefore, was unnecessary. One participant alluded to feeling protected, as all her cohabitees had been vaccinated: *“In my house, all of them are injected. Only I don’t get the vaccine. I rarely go out of my home. At this age (72 years old), it is very old. No need to inject.” (Female, 72 years old, history of stroke).*


*Fear of side effects*


Four of the eight interview participants expressed concern about the vaccine’s side effects, possibly related to the fact that all COVID-19 vaccines were rapidly developed and had been defined as emergency vaccines; indeed, the potential long-term effect of the vaccine is still unknown. In addition, there had been much negative speculation about potentially serious side effects on social media platforms, and this contributed to vaccine hesitancy. *“I don’t want to get the vaccine. I am not sure of its quality. I fear its side effects.” (Male, 70 years old, bipolar disorder).*


*Lack of information for decision-making*


Given that vaccines were rapidly developed and that many clinical trials were ongoing, there was anxiety that the evidence of their safety was not unequivocal. This lack of certainty contributed to confusion about decision-making around vaccination among the participants interviewed: *“I was interested in vaccination. But the information about the vaccines is not enough for me to decide to inject.” (Female, 72 years old, history of stroke).*


*Fear of death from the vaccine*


Most of the unvaccinated interview participants had one or more serious co-morbidities and long-term conditions. Although the Ministry of Public Health and the Centre for the Administration of the Situation Due to the Outbreak of the Communicable Disease Coronavirus 2019 announced that older people in Thailand should get the vaccine because they were classified as a high-risk group and often have other underlying medical conditions, some older people perceived that their many long-term conditions might result in their having serious side-effects of the COVID-19 vaccine, possibly resulting in death. *“I am afraid. Some people don’t have side effects after getting the vaccine, but some have side effects and die. I refuse to get the vaccine because I have many underlying medical conditions such as hypertension, stroke, glaucoma, diabetes, and high cholesterol. I fear the vaccine will lead me to not being able to see anything. Now I can only see with one eye. Many people die after the injection. I hear it from the news and other people talking. My grandchildren are too young. If I die, I won’t know my grandchild. So, I save myself by using a self-antigen test kit every week.” (Female, 68 years old, hypertension, stroke, glaucoma, diabetes, and dyslipidemia).*

## 4. Discussion

In our study, 92.4% of participants had already received or intended to receive the COVID-19 vaccine. This is consistent with other research among older people in the USA [14]. In our study, older people intended to receive the AstraZeneca vaccine because it was the first-choice vaccine (for first and second doses) arranged by the Thai government for older people. AstraZeneca phase II trials showed a good antibody response in those aged 70–80 years [3]. Most of the participants who intended to receive the COVID-19 vaccine had an underlying disease, particularly hypertension (41.98%), hyperlipidemia (26.60%), and diabetes (15.27%). Previous studies have found associations between hypertension and risk for severe disease and worse outcomes in older patients with COVID-19, namely that those with underlying hypertension are 2.3 times more likely to become seriously ill and 3.5 times more likely to die from COVID-19 compared to those without hypertension [23].

Our study found that perceived severity of COVID-19, perceived barriers, cues to action, and perceived benefits were statistically significant differences in the ‘intention’, ‘no intention’, and ‘not sure’ groups. This was supported by our qualitative findings, where vaccinated participants reflected on the severity of the COVID-19 pandemic, on their fear of death by COVID-19, and on the benefits of vaccination, including the prevention of infection and protection from serious illness if infected. Previous research found that perceived severity and perceived benefits were statistically significantly associated with decisions to get the COVID-19 vaccine [24,25], and cues to action were correlated to vaccine acceptance [25].

The predictors of participants’ intentions to get the COVID-19 vaccine (in the multinomial logistic regression) were perceived barriers to vaccination, perceived benefits, and health status. This result is consistent with previous research which has shown a higher rate of vaccination intention among older people with long-term conditions, who were more likely to accept the COVID-19 vaccine [18,25]. Another of the predictors of vaccination was its perceived benefits [18,25,26]. As the qualitative results from this study showed, participants reflected that it was: *“Better to inject than not inject*”. In addition, those who perceived significant barriers to access were less likely to accept the vaccine. Our participants had low levels of perceived barriers because those who were physically active could travel to vaccination centers, while those who were home-bound or bedridden had access to the proactive care service of home health care teams who visit the home to deliver the vaccination. We also found that participants chose to be vaccinated based on two main factors: the wish to continue with daily living and the wish to stay well. Our qualitative findings showed that reasons to be vaccinated included convenience for daily life and the wish to travel more freely. This was especially related to travel to other provinces, particularly during the height of the pandemic, when travelers needed to show evidence of vaccination status to be allowed to leave their home province. As findings from Switzerland have shown, older people have a willingness to be vaccinated against COVID-19 because of the possibility of return to their ‘normal’ life and because it relieves their anxiety; these factors constitute significant psycho-social benefits of the COVID-19 vaccine [27]. A study in Australia also showed that other reasons for the uptake of the vaccine are community protection [28,29,30] and domestic and international travel [26], and similar findings have also been identified in cross-sectional studies in a Thai clinic for older people in Bangkok [12] and in Switzerland [27]. Our participants explained how they had trust in the vaccines that the government provided and those endorsed by the medical profession. Similarly, a study of older Chinese people in Hong Kong showed they were willing to be vaccinated because they trusted the government to provide an effective vaccine, had confidence in the security of the vaccine, and saw vaccination as a civic responsibility [13].

For our unvaccinated group, the participants worried about vaccine side effects, particularly death. This is consistent with many previous studies that have found older people to lack confidence in the vaccine. Their concerns about vaccine safety [2,31,32,33,34,35,36], side effects or adverse effects [37], effectiveness, and long-term effects [2] have been identified. Therefore, trust in vaccine safety is a crucially influential factor influencing personal decisions around vaccination [38]. As older adults in Hong Kong lack trust and confidence in the vaccine, some perceive vaccines to be dangerous. They are worried about the safety and side effects of the vaccine. Moreover, they perceived that the vaccines were only effective for a short three-month period, with longer-term effects being unknown. Perceptions about toxicity have been revealed [39], with negative information about vaccines being received from various sources [36]. Therefore, a high level of willingness to receive vaccination among government and public health sectors should lead people to have confidence in the vaccine, along with the widespread deployment of information about the effectiveness of the vaccine [40,41,42]. In addition, this study found that some older people with long-term conditions deferred the decision to be vaccinated until the WHO specifically recommended that they should be. This finding is similar to a study among adults with long-term conditions in Saudi Arabia, where willingness to be vaccinated is low [40,41,42]. That may be because the government is faced with the spread of misinformation and a lack of strategy for building trust and confidence in the vaccine. Some participants pointed out that there is not enough information for them to make the decision. This may lead to uncertainty. Uncertainty is also related to the short development time of vaccines and concerns about long-term health consequences [30]. Therefore, the government should provide clear information on the COVID-19 vaccine to older people [37]. Healthcare providers can offer information and counseling about the COVID-19 vaccine in primary care [30], and trusted sources of information can play a role in increasing people’s vaccine acceptance [16]. The uptake of COVID-19 vaccination depends on the perception of the vaccine’s safety and efficacy, as well as communication strategy, especially at the community level, aimed at building trust; this should be communicated by policymakers [43].

### Strengths and Limitations

The quality of our study was enhanced through the use of a mixed-methods approach in which interviews were used to better understand responses from participants in the questionnaire. However, there were also some limitations. We collected data online; this may have limited access by some older people who didn’t use a smartphone or engage in social media, though data collection was also facilitated via face-to-face interaction. Additionally, over 50% of the 60+ Thai population are internet users [44]. We used convenience rather than probability sampling, which has implications for the representativeness and generalisability of the quantitative component for the wider over-60 population in Phuket and beyond. It was not possible to calculate the response rate and we did not glean any information about non-responders.

The cross-sectional nature of the research limits understanding of changes in attitudes or behaviors over time. The study was conducted during the second and third waves of the pandemic in Thailand, and it is likely that a current follow-up study might give different responses. This highlights the need for longitudinal research in the area of vaccine hesitancy and uptake to increase understanding of changes over time in response to the progression of pandemics and subsequent public health responses.

## 5. Conclusions

The results suggested that policies of intervention and campaigns addressing COVID-19 vaccination should employ strategies, such as the increased use of social and other popular media, to increase awareness of the potential benefits of vaccination while addressing perceptions around potential barriers to receiving the vaccine. In addition, the decision to be vaccinated or remain unvaccinated against COVID-19 is complex. In our study, participants who were from the older population were most concerned about disease prevention and protection, the convenience of going out, fear of contracting COVID-19, fear of death from COVID-19, and trust in the government-provided vaccine. In other words, in the unvaccinated group, key concerns related to the vaccine’s side effects and long-term health effects. Participants felt the result of the vaccine is uncertain. Because people had been exposed to information on both the advantages and disadvantages of vaccination, there is a need to rectify misinformation relating to the vaccination; one route to this might be via the use of social media, which could be used by the Ministry of Public Health. Moreover, the Thai government may provide a pragmatic strategy to foster the uptake of COVID-19 vaccination, which may help to motivate older people to receive vaccination in the future.

## Figures and Tables

**Table 1 ijerph-20-05919-t001:** Demographic characteristics of the participants and intention to be vaccinated (n = 406).

Variables	Total n (%)	No Intention n (%)	Intention n (%)	Not Sure n (%)	*p*
Sex Male Female	128 (31.5) 278 (68.5)	6 (1.5) 7 (1.7)	116 (28.6) 259 (63.8)	6 (1.5) 12 (2.9)	0.502
Age 60–69 yrs 70–79 yrs 80–89 yrs ≥90 yrs	245 (60.3) 124 (30.5) 33 (8.2) 4 (1.0)	5 (1.2) 5 (1.2) 2 (0.5) 1 (0.3)	230 (56.7) 114 (28.1) 28 (6.9) 3 (0.70)	10 (2.5) 5 (1.2) 3 (0.7) 0 (0.00)	0.05
	M = 68.8 SD = 6.7				
Education <Bachelor’s degree Bachelor’s degree Postgraduate degree	222 (54.7) 132 (32.5) 52 (12.8)	10 (2.5) 2 (0.5) 1 (0.3)	200 (49.3) 126 (31.0) 49 (12.1)	12 (3.0) 4 (1.0) 2 (0.5)	0.110
Occupation					0.204
Not working	143 (35.2)	9 (2.2)	127 (31.3)	7 (1.7)	
Working	131 (32.3)	2 (0.5)	121 (29.8)	6 (1.6)	
Retired	132 (32.5)	2 (0.5)	127 (31.3)	3 (0.7)	
Underlying disease No Yes	120 (29.6) 286 (70.4)	2 (0.3) 11 (2.7)	116 (28.6) 259 (63.8)	2 (0.5) 16 (3.9)	0.103
Perceived overall health Fair/Moderate Good	184 (45.3) 222 (54.7)	5 (1.2) 8 (2.0)	165 (40.7) 210 (51.7)	14 (3.4) 4 (1.0)	0.000 *
History of other vaccines received after 60 years old					0.051
No	151 (37.2)	9 (2.20)	6 (33.5)	6 (1.5)	
Yes	255 (62.8)	4 (1.0)	239 (58.9)	12 (3.0)	
Received information about the COVID-19 vaccine					0.189
No	15 (3.8)	0 (0.0)	13 (3.2)	2 (0.5)	
Yes	391 (96.2)	13 (3.2)	362 (89.2)	16 (3.9)	
◯ Sources of information (more than one answer)					0.189
Television	297 (73.2)	11 (2.7)	271	15	
Health care provider	169 (41.6)	2 (0.5)	161	6 (33.5)	
Line/Facebook	207 (51.0)	3 (0.7)	197 (48.5)	7 (1.7)	
Friends	138 (34.0)	3 (0.7)	129 (31.8)	6 (33.5)	
Family members	225 (55.4)	4 (1.0)	211 (52.0)	10 (2.5)	
Newspaper	24 (5.9)	1 (0.3)	21 (5.2)	2 (0.5)	
Other	14 (3.4)	1 (0.3)	12 (3.0)	1 (0.3)	

◯ percent of cases, * *p* < 0.05.

**Table 2 ijerph-20-05919-t002:** Mean, SD, mean rank, and Kruskal-Wallis test of perception of COVID-19 vaccine in the intention group to get the vaccine (n = 406).

Perception	No Intention to Be Vaccinated (n = 13)		Intention to Be Vaccinated (n = 375)		Not Sure (n = 18)		$x^{2}$ (*p*-Value)
	Mean(SD)	Mean Rank	Mean(SD)	Mean Rank	Mean(SD)	Mean Rank	
Perceived susceptibility	1.8(0.23)	131.7	2.4(0.05)	205.4	2.5(0.23)	215.5	5.2(0.075)
Perceived severity	3.3(0.31)	126.7	3.9(0.05)	205.2	4.1(0.16)	224.0	6.3(0.043 *)
Perceived barriers	2.0(0.23)	253.3	1.7(0.04)	196.8	2.4(0.17)	307.8	18.3(0.000 *)
Cues to action	1.3(0.22)	32.4	3.3(0.04)	213.3	2.6(0.17)	122.1	39.0(0.000 *)
Perceived benefits	3.0(0.31)	77.4	4.2(0.04)	212.8	3.5(0.13)	101.4	31.8(0.000 *)

Kruskal-Wallis test, * *p* < 0.05.

**Table 3 ijerph-20-05919-t003:** Pairwise comparison of the perception of the COVID-19 vaccine in the group intending to get the vaccine.

Pairwise Comparison in the Intention Group to Get the Vaccine	Test Statistics (Adjusted Significance)			
	Perceived Severity (*p* Value)	Perceived Barriers (*p* Value)	Cues to Action (*p* Value)	Perceived Benefits (*p* Value)
No intention-Not sure	−78.5 (0.051)	56.8 (0.246)	−89.8 (0.106)	−24.1 (1.00)
No intention-Intention	−97.4 (0.066)	−111.0 (<0.001 *)	−180.9 (<0.001 *)	−135.4 (<0.001 *)
Not sure-Intention	−18.8 (1.000)	−54.3 (0.593)	91.2 (0.004 *)	111.3 (<0.001 *)

Adjusted by the Bonferroni correction for multiple tests, * *p* < 0.05.

**Table 4 ijerph-20-05919-t004:** Multinomial regression analysis predicting intention for COVID-19 vaccine uptake.

Variables	To Receive a Vaccine Compared with No Intention to Receive a Vaccine		Not Sure Compared to Having No Intention to Receive a Vaccine	
	AdjOR (95% CI)	*p*	AdjOR (95% CI)	*p*
Age	0.97 (0.90–1.05)	0.451	1.07 (0.97–1.19)	0.191
Perceived barriers	0.32 (0.17–0.59)	<0.001	0.63 (0.26–1.15)	0.299
Perceived benefits	2.65 (1.49–4.71)	<0.001	0.70 (0.35–1.42)	0.323
Health status: moderate	Ref		Ref	
Health status: not good	0.10 (0.02–0.49)	0.005	0.00 (0.00–0.00)	
Health status: good	3.51 (1.01–12.12)	0.048	5.47 (1.02–29.30)	0.047

**Table 5 ijerph-20-05919-t005:** Characteristics of interview participants.

No.	Age	Male (M)/ Female (F)	Medical History/ Co-Morbidities	History of Flu Vaccine	COVID-19 Vaccine	Decision-Making around COVID-19 Vaccination	Who Influenced Decision around COVID-19 Vaccination
1	79	M	Colon cancer, BPH	Yes	Yes	Hesitated	Family
2	77	F	Hypertension Dyslipidemia	Yes	Yes	Hesitated	Family
3	76	F	Hypertension Hyperthyroidism	No	Yes	Immediately vaccinated	Self
4	82	F	Diabetes Hypertension	Yes	Yes	Immediately vaccinated	Self
5	69	F	Non	No	No	Refused	Self
6	70	M	Bipolar Disorder	No	No	Refused	Self
7	65	F	Hypertension Dyslipidemia	No	Yes	Hesitated	Self
8	79	F	Diabetes Hypertension	Yes	Yes	Immediately vaccinated	Family
9	77	F	Diabetes Hypertension	Yes	Yes	Hesitated	Self
10	81	F	Hypertension Dyslipidemia	Yes	Yes	Immediately vaccinated	Self
11	60	F	Hypertension	Yes	Yes	Immediately vaccinated	Self
12	89	F	GERD	No	Yes	Immediately vaccinated	Self
13	68	M	None	Yes	Yes	Immediately vaccinated	Self
14	62	F	None	Yes	Yes	Immediately vaccinated	Self
15	69	F	Allergic rhinitis	No	Yes	Immediately vaccinated	Self
16	60	F	None	No	Yes	Hesitated	Self
17	70	F	Breast cancer	No	Yes	Immediately vaccinated	Family
18	74	F	Hypertension	No	Yes	Hesitated	Family
19	68	F	Hypertension	No	Yes	Immediately vaccinated	Family
20	69	F	Diabetes Hypertension Dyslipidemia	Yes	Yes	Immediately vaccinated	Self
21	78	F	Hypertension Dyslipidemia	No	Yes	Immediately vaccinated	Family
22	62	F	GERD	Yes	Yes	Immediately vaccinated	Self
23	61	F	Dyslipidemia	Yes	Yes	Immediately vaccinated	Self
24	65	M	Gout	Yes	Yes	Immediately vaccinated	Self
25	63	F	Dyslipidemia, Hyperthyroid	Yes	Yes	Immediately vaccinated	Self
26	74	F	Hypertension	No	No	Refused	Self
27	68	F	Stroke Diabetes Hypertension Dyslipidemia Glaucoma	No	No	Refused	Self
28	79	F	Diabetes Hypertension	No	No	Refused	Self
29	80	F	Stroke	No	No	Refused	Self
30	84	M	Diabetes Hypertension	No	No	Refused	Self
31	78	M	Non	No	No	Refused	Self
32	60	F	Knee pain	No	Yes	Immediately vaccinated	Self
33	68	F	Hypertension Dyslipidemia	Yes	Yes	Immediately vaccinated	Self
34	61	F	Diabetes Hypertension	No	Yes	Immediately vaccinated	Self
35	64	F	Hypertension	Yes	Yes	Immediately vaccinated	Self
36	62	M	Hypertension	Yes	Yes	Immediately vaccinated	Self

BPH: Benign prostatic hyperplasia, GERD: Gastroesophageal reflux disease.

## Data Availability

The data presented in this study are available on reasonable request from the corresponding author. The data are not publicly available for reasons of confidentiality.

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
