# Peer review of "Perceptions and Intentions around Uptake of the COVID-19 Vaccination among Older People: A Mixed-Methods Study in Phuket Province, Thailand"

_ijerph, 2023, doi:10.3390/ijerph20115919_

Round 1

Reviewer 1 Report

1.      What does the meaning of “COVID-19 vaccines are one of many important public health interventions to control the spread of infection and contribute to achieving herd immunity and reduce the risk of severe illness”? Vaccines or natural infections are the only way to herd immunity.

2.      Some mistakes related to grammar

3.      Are vaccinations for COVID-19 in Thailand is doing free of cost or paid?

4.      Are people showing indecision toward all types of vaccines or some particular vaccine for COVID-19.

5.      Suggest some alternate of vaccine

6.      Are 406 participant numbers sufficient or not for this study as per the statistician?

7.      The review of the present is more related to the statistician, so they can be the better judge.

Author Response

Please find attached the responses to your very helpful review and that of the other reviewer too.

Key changes have also been highlighted in the main body of the text.

With many thanks,

Ros

Reviewer 2 Report

Congratulation to all authors for an interesting mixed-methods research manuscript which is crucial for both practitioners, policymakers and researchers.  Overall looks like nicely drafted a mixed-methods manuscript but there are a few unclarities that should be addressed prior to approval for the publication:

1.     Abstract section: Please insert the Multinomial regression analysis revealed significant variables OR and CI for your readers.

2.     Para 16 8participants' = make space and write in word  i.e. four, eight.

3.     Provide a pragmatic strategy to  foster the uptake of COVID-19 vaccine in Phuket elderly community.

Introduction: Nicely narrated but at the beginning if you summarises the few COVID-19 vaccines so far in the globe and how many vaccines are available in Thailand which would be better concept of this context.

Second why Phuket elderly why not other parts of the elderly community in Thailand? Give few inequalities issue related indicators among this population.

Materials and methods

·       Numerical is incorrect i.e 2.2.1 and 1 data collection. So make this consistent and correct.

·       Para 60 insert one mixed-methods research book reference which helps to sing post of the sequential explanatory design.

·       I could not understand why para 63-65 : “Sub-sequently, semi-structured individual interviews were conducted with a subsample of the participants who completed the online survey to quantify the different views on COVID-19 vaccine acceptability and refusal and to explore the reasons behind these views”. Keep under the respective headings not under the study design.

·       How did you determine 338? Which is not clear please provide with detail calculation. What was the reason behind increased of 20% sample size instead of non-response rate calculation. Make this clear.

·       Para 119 – 121 font size is inconsistent make this consistent.

Overall reading of this section found confusion while researcher is presenting the mixed-methods research methods. So, I would suggest present both methods section under each headings that helps your writing in clear and coherent otherwise if you present quantitative methods with sub-headings and qualitative under the single headings which is not the pragmatic philosophy.  

·       How did you select the qualitative participants? Are the sub-sample of quantitative survey or  are totally new participants. It would be great to brief socio-demographic characteristic of those 36 participants.

Results

·       Para 146 provide space of 1 i.e. Table 1Demographic.

·       Make Table 1 reader friendly i.e. education category which is not appropriate for reading and also recategorize those 0 cell values i.e. Diploma, employee, agriculture, Others, Chronic kidney disease, Fair/Poor, No.

·        Make correct of para 147 which is not appropriate table format which you can make repeat header row.

·       Make reader friendly of Table 2 too and present SD consistency with two decimal point i.e. (0.23).

·       Make clear and correct way of writing Table 3 para 171 to 178 i.e. p value should not .000 this should be <0.001 and do not make confusion on writing .05 better to remove those under this paras. Make all below tables 3 and 4 too writing 0. And if 0.000 values then write in <0.001 form in both narration and tables.

·       Para 196 is confusion where in the methodology stated 36 interviews for the qualitative data but in para 196 this is 28. So make this correct and present key socio-demographic information under the qualitative participants.

·       The way of mixed methods finding presentation is crucial and in the methods section researchers had mentioned the explanatory methods which is very crucial if the findings come under the respective quantitative findings that may give better sense for the mixed methods research. So I would suggest you make research like really a mixed-methods not multi-methods and one approach explain to another.

Discussion

·       Discussion should integrate the mixed methods findings and present accordingly where many emergent themes are presented in qualitative findings which are very limited discussion under this section.

Author Response

(The authors gave the same response as above.)

Round 2

Reviewer 1 Report

All suggested change has been done

Author Response

With many thanks for confirming you were happy with the changes.